# Factors Related to the Progression of Clinically Isolated Syndrome to Multiple Sclerosis: A Retrospective Study in Lithuania

**DOI:** 10.3390/medicina58091178

**Published:** 2022-08-30

**Authors:** Renata Balnytė, Vaidas Matijošaitis, Ieva Čelpačenko, Miglė Malciūtė, Radvilė Stankevičiūtė, Ovidijus Laucius

**Affiliations:** Department of Neurology, Lithuanian University of Health Sciences Medical Academy, A. Mickevičiaus g.9, LT-44307 Kaunas, Lithuania

**Keywords:** multiple sclerosis, clinically isolated syndrome, conversion

## Abstract

*Background and Objectives*: Multiple sclerosis (MS) is a demyelinating disease which usually manifests as clinically isolated syndrome (CIS). Approximately 70% of patients with CIS progress to MS. Therefore, there is a pressing need to identify the most accurate predictive factors of CIS developing into MS, some of which could be a clear clinical phenotype of early MS as well as lesions in magnetic resonance imaging (MRI), pathological findings in cerebrospinal fluid (CSF) and evoked potentials (EP) tests. The problem is of outstanding importance since early MS diagnosis and treatment prevents long-term disability. The aim of our study is to analyze the factors that could influence the progression of CIS to MS. *Materials and Methods*: This study is a retrospective data analysis which included patients with their primary CIS diagnosis between 1st January 2015 and 1st January 2020. The prevalence and predictive value of clinical symptoms, MRI lesions, pathological CSF and EP findings were evaluated in accordance with the final diagnosis and compared between the sexes and age groups. *Results*: Out of 138 CIS patients, 49 (35.5%) patients progressed to MS. MS patients were more likely to have a diminished sense of vibration and proprioception (χ^2^ = 9.033, *p* = 0.003) as well as spinal cord MRI lesions (χ^2^ = 7.209, *p* = 0.007) in comparison with the non-MS group. Positive oligoclonal bands (OCBs) in CSF (χ^2^ = 34.859, *p* ≤ 0.001) and pathological brainstem auditory evoked potential (BAEP) test findings (χ^2^ = 10.924, *p* ≤ 0.001) were more prevalent in the MS group. Diminished sense of vibration and proprioception increased the risk for developing MS by 13 times (*p* = 0.028), whereas positive OCBs in CSF increased the risk by 100 times (*p* < 0.001). MS patients that were older than 50 years were more likely to exhibit positive Babinski’s reflex (χ^2^ = 6.993, *p* = 0.03), decreased muscle strength (χ^2^ = 13.481, *p* = 0.001), ataxia (χ^2^ = 8.135, *p* = 0.017), and diminished sense of vibration and proprioception (χ^2^ = 7.918, *p* = 0.019) in comparison with both younger age groups. *Conclusions*: Diminished sense of vibration and proprioception, spinal cord MRI lesions, positive OCBs and pathological BAEP test findings were more common among patients that developed MS. Diminished sense of vibration and proprioception along with positive CSF OCBs are predictors of CIS progressing to MS. Older patients that develop MS have more symptoms in general, such as positive Babinski’s reflex, decreased muscle strength, ataxia, and diminished sense of vibration and proprioception.

## 1. Introduction

Multiple sclerosis (MS) is a chronic demyelinating central nervous system (CNS) disease which commonly manifests as clinically isolated syndrome (CIS) [1], which lasts for at least 24 h and can affect the brain, brainstem, optic nerve, and spinal cord [2,3]. The broad spectrum of MS lesion locations makes its presentation heterogeneous—the disease can affect anything from vision to coordination to sphincter control [4]. Therefore, it proposes a great challenge for clinicians to associate these clinical features with MS.

The latest 2017 McDonald Criteria for the Diagnosis of Multiple Sclerosis is expected to speed the diagnosis of MS and make it less complex [5]. The key changes that were made in the 2017 McDonald Criteria were: positive oligoclonal bands (OCBs) in cerebrospinal fluid can substitute for dissemination in time (DIT); both asymptomatic and symptomatic MRI lesions can be considered as dissemination in space (DIS) or time; and cortical lesions have been added as determining MRI criteria for DIS.

MS mostly affects young adults, and the incidence rate has increased over the years [6]. It is of outstanding importance to determine a clear phenotype of early MS that could help establish a more efficient diagnostic process. Moreover, there is a necessity to identify factors of the progression of CIS to MS. Some of them could be pathological findings in magnetic resonance imaging (MRI), cerebrospinal fluid (CSF) and evoked potentials (EP) that were not included in the latest 2017 McDonald Criteria. It is crucial to establish clinical features and more accurate MRI, CSF, and EP tests findings of early MS, since delayed MS diagnosis increases the risk of disability [7]. Furthermore, early pharmacological treatment and rehabilitation therapy improves outcomes in patients with early MS; constraint-induced movement therapy (CIMT) and medication treatment seems to be effective in improving upper limb dexterity in MS patients [8,9].

Lithuania reports higher MS incidence rates (78 in 100,000 people) compared to other countries [10]. Therefore, our aim was to establish clear clinical symptoms and signs, such as MRI lesions and pathological findings in CSF and EP testing, as possible factors for the progression of CIS to MS in the Lithuanian population, as there are no other studies investigating this topic in Lithuania.

## 2. Materials and Methods

This study is a retrospective data analysis of medical records gathered at the public Hospital of Lithuanian University of Health Sciences (LUHS) Kaunas Clinics Department of Neurology between 1 January 2015 and 1 January 2020. Ethical approval was obtained from the local LUHS Department of Bioethics (Approval No. BEC-MF-30).

Our study included patients with primary CIS diagnoses confirmed by board-certified neurologists. Patients were divided into 2 groups in accordance with the final diagnosis (Multiple sclerosis (MS) and non-Multiple sclerosis (non-MS) groups) and into 3 age groups (18–30 (Group 1), 30–50 (Group 2), >50 years old (Group 3)). In this study, CIS patients’ ages ranged broadly from 18 to 73 years, so it was difficult to evaluate the data without dividing patients into three separate groups, since some findings (MRI detected lesions or pathological Brainstem Electric Response Audiometry test findings) are related to the patient’s age [11]. Moreover, we aimed to investigate correlations between patients’ age and their symptoms as well as MRI, CSF, and EP findings characteristic to a certain age group. The incidence rate of CIS progressing to MS was analyzed between the groups and the sexes. Regarding the retrospective nature of the study, clinical variables were acquired from medical documentation, such as anamnesis and neurological examination performed by neurologists in LUHS Kaunas Clinics. The prevalence of clinical characteristics and symptoms was determined (fatigue, generalized weakness, pain, decreased muscle strength, abnormal reflexes, muscle tone abnormalities, vertigo, cranial nerves dysfunction, pathological reflexes: Babinski‘s reflex, Rossolimo‘s reflex, diminished sense of vibration, proprioception and superficial sensations, ataxia, imbalance, and urinary incontinence and retention) between MS/non-MS groups and age groups. All the clinical findings and laboratory data (OGBs status, MRI findings, and data of visual evoked potentials (VEPs, BEAP)) were reviewed retrospectively from the medical records. Lumbar puncture and cerebrospinal fluid analysis were performed at the time of CIS diagnosis. All imaging studies were conducted with a 1.5-T MR scanner (MAGNETOM Avanto, Siemens, Erlangen, Germany) with a standard head coil. The registration of VEPs and BEAP was completed by the Evoked Potential Navigating System (Bio-Logic System Corp., Mundelein, IL, USA). Matched CSF and plasma samples were analyzed using isoelectric focusing and IgG-specific immunofixation to test for the presence of intrathecal-specific OCBs and compared directly with the serum samples. OCBs were defined as positive if more than 2 bands were present in the CSF but absent in the corresponding blood serum.

The frequency of specific and unspecific brain and spinal cord MRI lesions, IgG levels, positive OCBs in CSF and lesions detected by BAEP and VEP tests were compared between the groups. MRI lesions were analyzed in accordance with their localization (juxtacortical, periventricular, infratentorial).

### 2.1. Patient Inclusion and Exclusion Criteria

Included patients were adults (aged ≥ 18 years) diagnosed with CIS in accordance with 2017 McDonald Criteria for the Diagnosis of Multiple Sclerosis. The CIS diagnosis was determined by board-certified neurologists in LUHS Kaunas Clinics, and a broad differential diagnosis was performed. Patients were tested for neuroinfections, such as neurosyphyllis, neuroboreliosis, and tick-borne encephalitis. Neuroimaging was performed to differentiate between demyelinating and non-demyelinating lesions. MS diagnosis was confirmed by 3 board-certified neurologists. Medical records between 1 January 2015 and 1 January 2020 of included CIS patients were analyzed for the development of CIS to MS in the given period. Insufficient MRI data was considered to be the main criterion for exclusion.

### 2.2. Statistical Analysis

Statistical analysis was performed using SPSS (Statistical Package for the Social Sciences) version 24.0. Chi-squared test was employed for comparison between categorical variables. The association between two quantitative variables was determined through the Spearman correlation coefficient. All quantitative variables were distributed normally with the exception of the sum of related neurological signs and symptoms. The Mann–Whitney U test and the independent samples *t*-test were used for comparison between the groups. The power of the study was estimated using one sample *t*-test. Multivariate logistic regression was employed for determining potential MS risk factors. The relative risk ratio was estimated and examined with the Wald X^2^ test at 95% confidence limits. Factors related to subsequent MS diagnosis were selected using univariate analysis. Variables that had a significant result in univariate testing were selected as candidates for the multivariate analysis. The selection was based on the Wald test from logistic regression and the cut-off point of *p*-value was 0.05. Results were interpreted as statistically significant when *p*-value < 0.05.

## 3. Results

### 3.1. Demographic Characteristics

A total of 169 CIS cases were reviewed, although 31 were excluded due to insufficient data. Thus, 138 CIS patients were enrolled in the study: 92 (64.5%) females and 46 (35.5%) males, mean age 43.83 (±13.49), age ranged between 18 and 74. Of the 138 subjects enrolled, 49 (35.5%) patients (28 (57.1%) females and 21 (42.9%) males, mean age 42.59 years (±14.88), age ranged between 19 and 74) fulfilled the 2017 McDonald’s criteria for MS by dissemination in space and time and comprised the MS group. The remaining 89 patients (25 (28.1%) males and 64 (71.9%) females, mean age 44.52 years (±12.69, age ranged between 18 and 72) either remained with CIS diagnosis or were diagnosed with other diseases (Devic’s disease, neuroborreliosis), and comprised the non-MS group.

MS patients were subdivided into the following age groups. Group 1: 11 (22.4%) patients 18–30 years (mean age 24.36 ± 3.80), group 2: 19 (38.8%) patients age 31–50 years (mean age 37.16 ± 6.17) and group 3: 19 (37.1%) patients age 50+ (mean age 58.58 ± 6.38).

The prevalence of developing MS did not differ between the sexes or age groups. The demographic characteristics of the study group are shown in Table 1.

### 3.2. Clinical Characteristics

MS patients were more likely to have a diminished sense of vibration and proprioception (*n* = 20, 41.7%) in comparison with the non-MS group (*n* = 16, 18.0%), (χ^2^ = 9.033, *p* = 0.003). There were no other statistically significant clinical differences between the MS and non-MS groups and between the sexes. Results are shown in Table 2.

MS patients that were older than 50 years were more likely to exhibit positive Babinski’s reflex (χ^2^ = 6.993, *p* = 0.03), decreased muscle strength (χ^2^ = 13.481, *p* = 0.001), ataxia (χ^2^ = 8.135, *p* = 0.017), and diminished sense of vibration and proprioception (χ^2^ = 7.918, *p* = 0.019) in comparison with both younger age groups. Results are displayed in Table 3.

Moreover, moderate correlation was found between age and the sum of related neurological symptoms and signs in the MS group (*r*_s_ = 0.419, *p* = 0.03). Statistically significant differences between sums of related neurological symptoms and age groups were obtained: group 3 with the median of 6.00 symptoms (2–13) had twice as many symptoms as patients in group 1 (median = 3.00 (0–6)), U = 10.519, *p* = 0.005.

### 3.3. Magnetic Resonance Imaging

Results of magnetic resonance imaging were distributed in the following manner: 86 (63.31%) patients had MS-specific MRI lesions, 12 (8.69%) had MRI lesions indicative of other diseases and 40 (28.98%) had unspecified MRI lesions. The prevalence of MS-specific MRI lesions did not differ between MS and non-MS patients (χ^2^ = 2.445, *p* = 0.118). Nonetheless, unspecified MRI lesions were more common among non-MS patients (χ^2^ = 4.328, *p* = 0.037). Patients that developed MS were more likely to have spinal cord MRI lesions compared with the non-MS patients (χ^2^ = 7.209, *p* = 0.007). Results are displayed in Table 4.

No differences have been found between age groups in relation to spinal MRI lesions among MS patients (χ^2^ = 0.844, *p* = 0.656). In addition, the prevalence of different lesion locations in MRI did not differ between age groups (juxtacortical χ^2^ = 1.473, *p* = 0.479; periventricular χ^2^ = 0.619, *p* = 0.734; infratentorial χ^2^ = 0.760, *p* = 0.684).

### 3.4. Cerebrospinal Fluid

Positive OCBs were demonstrated in 40 out of 108 (37%) CIS patients. Positive OCBs in CSF were more prevalent in the MS group (χ^2^ = 34.859, *p* < 0.001).

Significant differences in IgG levels in CSF were found between the groups: the median of IgG levels in the MS group was 43.91 mg/L (8.42, 95.60) and the non-MS group median was 36.51 mg/L, (8.82,117), (*p* = 0.02). These results are shown in Figure 1.

Pleocytosis was demonstrated among 48 of 120 (40%) CIS patients, although prevalence of pleocytosis did not differ between MS and non-MS groups (χ^2^ = 0.739, *p* = 0.390) (Appendix A).

Moderate positive correlation was found between IgG levels in CSF and protein levels in CSF (r_s_ = 0.551, *p* = 0.002), as well as between IgG levels in CSF and white blood cell count in CSF (*r*_s_ = 0.399, *p* = 0.032) (Appendix A).

### 3.5. Evoked Potentials

Abnormal BAEP was found in 29 subjects (28.2%). Pathological BAEP test findings were more frequent in the MS group (χ^2^ = 10.924, *p* ≤ 0.001) (Figure 2).

However, no relation was determined between lesions detected by VEP test and MS diagnosis (χ^2^ = 2.210, *p* = 0.137). Although, it was assessed that pathological VEP test results were more common among MS patients older than 50 years (χ^2^ = 6.089, *p* = 0.048).

### 3.6. Predictive Models

Multivariate statistical strategy was implemented to quantify the predictive power of variables. Test sensitivity—81.3%, specificity—84.2%, overall—82.4%, and R^2^—0.466. Diminished sense of vibration and proprioception increased the risk of progression to MS by 13 times (OR 13.059, 95% CI (1.300–131.169), B 2.569, S.E. 1.177, *p* = 0.029), whereas positive OCBs in CSF increased the risk 100 times (OR 100.253, 95% CI (6.983–1439.317; B 4.608, S.E.1.359, *p* < 0.001). According to this predictive model, pathological BAEP findings (*p* = 0.598), higher levels of IgG in CSF (*p* = 0.138), and specific MRI lesions (p=0.055) had no predictive value for the progression of CIS to MS.

## 4. Discussion

This is the first Lithuanian study which investigates the progression of CIS to MS, which aimed to evaluate the clinical and paraclinical features of early MS in the Lithuanian population. Only 35.5% of included patients progressed to MS between 2015 and 2020. In 2013, an Italian study showed that at 2-year follow-up, 57% of CIS patients had developed MS according to McDonald criteria, 67% at 3 years and 75% at 4-year follow-up [3]. Given that, more MS cases could have been observed with longer follow-up periods.

According to our results, conversion rate did not differ between sexes nor age groups. Previous research indicates that females are 1.2 times more likely to develop MS as a consequence of CIS in comparison with males [12]. Thus, it could be hypothesized that our sample size was not big enough to observe such a difference. Moreover, younger patients seem to be more susceptible to developing MS following CIS [13,14,15,16]. Our results could have been affected due to the relatively low number of young participants.

The prevalence of symptoms was similar in both MS and non-MS groups. Only diminished sense of vibration and proprioception was found to be more prevalent in the MS group. This was also confirmed by multivariate logistic regression, as impaired deep sensation was associated with a 13 times higher probability of developing MS. Previous studies have shown that cerebellar dysfunction [16,17], brainstem CIS [16], bladder control problems [17,18] and sensory involvement [19] could predict the progression of CIS to MS. Older MS patients had more symptoms in general: they were more often observed to have positive Babinski’s reflex, ataxia, decreased muscle strength, diminished sense of vibration and proprioception. Thus, it could be hypothesized that various other conditions could have been present among the older population, or MS could have been diagnosed in a more advanced stage. Even though some of the results seem to be consistent with available literature, it could be assumed that neurological examination alone is not as informative when evaluating the progression of CIS to MS.

According to the newest McDonald criteria, MS can be diagnosed when dissemination in space (DIS) and time (DIT) is confirmed. DIS is proven when typical lesions are found in at least two of the following regions: juxtacortical, cortical, periventricular, infratentorial and spinal cord. DIT, however, could be established by experiencing another clinical attack, having positive CSF oligoclonal bands (OCBs) or having both Gadolinium-enhanced and non-enhanced lesions in MRI [5]. MS-specific MRI lesions are usually associated with development of MS [20,21]. According to our results, lesions specific to MS occurred at the same rate both in the MS and non-MS group, however, non-specific MRI lesions were observed more often among the non-MS group. Our study was limited due to its retrospective nature as we could only access the description of the MRI scan. Therefore, two subjective interpretations had to be performed: radiological evaluation and interpretation of MRI description by the authors. MRI lesions were divided into three groups: MS-specific lesions; lesions indicative of other diseases and unspecified lesions. Unspecified MRI lesions had to be differentiated between demyelinating, angiopathic or vasculitic lesions. A patient could have had either one type of lesion or all three of them. The majority of the patients included in the study (*n* = 86) had MS-specific MRI lesions; however, only 40 patients were observed to have unspecified MRI lesions. The fact that most patients had MS-specific MRI lesions but only 35% of the patients developed MS brought down the predictive value of MS-specific MRI lesions. Furthermore, MRI brain lesion evaluation in Lithuania is not defined by a specific protocol. As reported in our results, only spinal cord lesions were more prevalent in the MS group, even though evidence exists that periventricular, juxtacortical, [13] infratentorial [21], and spinal cord [22] lesions could predict the development of MS. This suggests that obtaining a spinal cord MRI could aid those whose diagnosis remains unclear.

Positive CSF OCBs is a well-established predictor of CIS progressing to MS [13,16,20,23,24]. Our results confirmed that 71.4% of MS patients had positive OCBs and the predictive model showed that patients with positive OCBs were 100 times more likely to develop MS. However, 10 patients who remained undiagnosed with MS were also OCB-positive. According to the available literature, CSF OCBs’ specificity to MS varies from 61–93%, ranking lowest when differentiating between other inflammatory CNS diseases [25], therefore, other pathologies should be considered, even with OCB-positive patients. MS patients presented higher IgG levels in CSF; it also positively correlated with protein levels and white blood cell count in CSF, which complies with previous publications [24].

Detection of prolonged VEP latency increases specificity of predicting a second attack in CIS patients, however, it is still not included in the MS diagnostic criteria [5]. Our study did not find prolonged VEP latency to be more common among patients that were diagnosed with MS. This could be due to the relatively small patient sample. On the contrary, impairment in BAEP results was more prevalent in the MS conversion group. An explanation for that could be the relatively old patient sample with a mean age of 43.83 (±13.49), as various comorbidities could have affected the results. It has been shown that multimodal evoked potentials (VEP, BAEP and somatosensory evoked potentials) could predict the development of MS [20] and disability progression [26] in CIS patients. Therefore, a combination of EP tests could be considered as an addition to MRI when evaluating CIS patients.

Important limitations include its retrospective nature, thus, having a prospective trial with an even bigger sample size could verify our knowledge about the features of early MS in Lithuanian patients. Moreover, anthropometric data and comorbidities could be added to the possible development factors. Moreover, our study only includes data from 1 January 2015 to 1 January 2020, which do not reflect the recent COVID-19 pandemic and its possible influence on an increased number of MS cases. The follow-up of our study was finalized before the COVID-19 pandemic, so we were not able to analyze COVID-19 disease as a conversion factor. However, various indicators were shown to be more prevalent among those who developed MS and some of them predict the development of MS. To our knowledge, no other trials regarding the Lithuanian population of CIS patients and their progression to MS have been performed. This study contains data from a 5-year span and gives a wide understanding of what are the most important features when determining if a CIS patient should be monitored even more closely in regard of developing another demyelinating attack. Moreover, certain prognostic factors of the progression of CIS to MS could aid in a more efficient diagnostic process and earlier prescription of disease-modifying therapy.

## 5. Conclusions

Diminished sense of vibration and proprioception, spinal cord MRI lesions, positive OCBs and pathological BAEP test findings are more common among CIS patients that later develop MS. Diminished sense of vibration and proprioception along with positive CSF OCBs are predictors of the progression of CIS to MS. Older patients that develop MS have more symptoms in general, such as positive Babinski’s reflex, decreased muscle strength, ataxia and a diminished sense of vibration and proprioception.

## Figures and Tables

**Figure 1 medicina-58-01178-f001:**
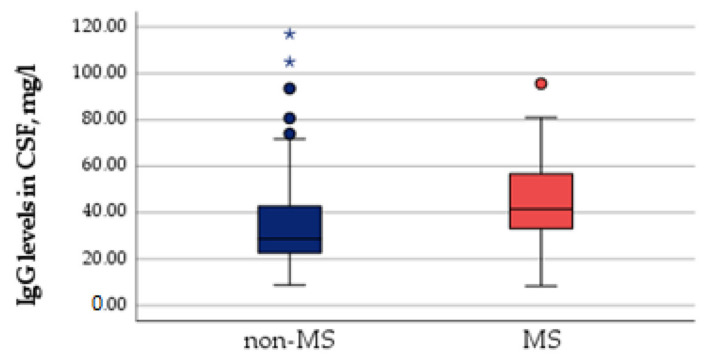
IgG levels in cerebrospinal fluid and conversion to MS. ***** Statistically significant differences are expressed in the bold type (*p* < 0.05).

**Figure 2 medicina-58-01178-f002:**
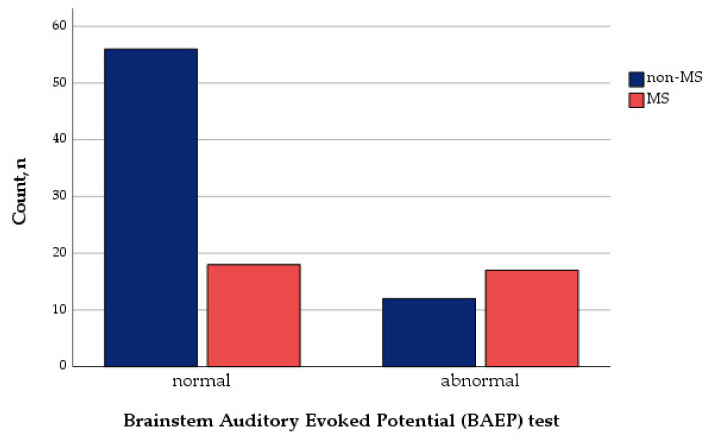
Brainstem Auditory Evoked Potential (BAEP) test and development of MS.

**Table 1 medicina-58-01178-t001:** Characteristics of the patients included in the study.

Characteristics	MS Patients, *n* (% within MS)	Non-MS Patients, *n* (% within Non-MS)	MS Patients vs. Non-MS Patients (*p*-Value)
Male	21 (42.9)	25 (28.1)	0.78
Female	28 (57.1)	64 (71.9)	
Group 1: 18–30	11 (22.4)	17 (19.1)	0.823
Group 2: 31–50	19 (38.8)	39 (43.8)	
Group 3: 50+	19 (37.1)	33 (38.8)	

Abbrevations: MS patients- Multiple sclerosis patients; Non-MS patients- non-Multiple sclerosis patients; Statistically significant differences are expressed in the bold type (*p* < 0.05).

**Table 2 medicina-58-01178-t002:** Clinical characteristics of the study group.

Clinical Characteristics	MS Patients, *n* (%)/Non-MS Patients, *n* (%)	MS Patients vs. Non-MS Patients (*p*-Value)
Fatigue	4 (8.2)/13 (14.6)	0.27
Generalized weakness	5 (10.2)/9 (10.1)	0.986
Pain	11 (22.4)/32 (36.0)	0.101
Vertigo	14 (28.6)/37 (41.6)	0.130
Cranial nerves dysfunction	29 (59.2)/47 (52.8)	0.471
Babinski’s reflex	22 (44.9)/30 (33.7)	0.194
Rossolimo’s reflex	12 (24.5)/20 (22.5)	0.788
Decreased muscle strength	21 (42.9)/27 (30.3)	0.139
Abnormal deep tendon reflexes	33 (67.3)/54 (60.7)	0.437
Muscle tone abnormalities	8 (16.3)/7 (7.9)	0.126
Diminished sense of vibration and proprioception	20 (41.7)/16 (18.0)	**0.003**
Diminished sense of superficial sensation	17 (35.4)/30 (33.7)	0.841
Ataxia	21 (43.8)/39 (43.8)	0.994
Imbalance	21 (43.8)/37 (41.6)	0.806
Urinary incontinence	6 (12.5)/5 (5.6)	0.193
Urinary retention	2 (4.2)/2 (2.2)	0.612

Abbrevations: MS patients- Multiple sclerosis patients; Non-MS patients- non-Multiple sclerosis patients; Statistically significant differences are expressed in the bold type (*p* < 0.05).

**Table 3 medicina-58-01178-t003:** Distribution of clinical features in accordance with age groups.

	Group 1: 18–30 y.o., *n* (%)	Group 2: 31–50 y.o., *n* (%)	Group 3: 50+ y.o., *n* (%)	*p*-Value
Positive Babinski‘s reflex	3 (27.3)	6 (31.6)	13 (68.4)	**0.030**
Decreased muscle strength	1 (9.1)	6 (31.6)	14 (73.7)	**0.001**
Ataxia	2 (20)	6 (31.6)	13 (68.4)	**0.017**
Diminished sense of vibration and proprioception	1 (9.1)	7 (36.8)	12 (63.2)	**0.019**

y.o—years old; Statistically significant differences are expressed in the bold type (*p* < 0.05).

**Table 4 medicina-58-01178-t004:** MRI findings.

	Non-MS Patients, *n* (% within Non-MS Group)	MS Patients, *n* (% within MS Group)	*p*-Value
MS-specific MRI lesions	51 (58)	35 (71.4)	0.118
Unspecified MRI lesions	31 (35.2)	9 (18.4)	**0.037**
MRI spinal cord lesions (+)	12 (41.4)	16 (80)	**0.007**
MRI spinal cord lesions (−)	17 (58.6)	4 (20)	

Statistically significant differences are expressed in the bold type (*p* < 0.05).

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
