# Peer review of "Factors Related to the Progression of Clinically Isolated Syndrome to Multiple Sclerosis: A Retrospective Study in Lithuania"

_medicina, 2022, doi:10.3390/medicina58091178_

Round 1
Reviewer 1 Report (Previous Reviewer 2)
1. Abstract.
a. The methods part of the abstract should mention what was analyzed in the neuroimaging study? Labs? Electrodiagnostic studies? Only clinical symptoms?
b. It is advised to have a complete description of CSF.
c. Could the objective of the study be more precise in the abstract?
2. Methods.
a. Why were the patients divided into three age groups regarding the range of aging chosen?
b. How was calculated the power of the study?
c. Could the authors upload the Spearman correlation graphics simulated by the statistical program as supplementary material?
d. Could the authors provide the spreadsheet database as supplementary material or publish it in Mendeley Data?
e. What were the model and variables chosen for the logistic regression?
f. We have three groups of individuals. Why did the authors perform ANOVA?
g. All the patients performed the same laboratory exams, neuroimaging, and electrodiagnostic studies?
3. Results
a. Could the authors upload a table with other diagnoses besides MS and percentages after the first CIS presentation? This information can be uploaded as supplementary material and would highly impact the quality of the manuscript.
b. The table with results of logistic regression should include only significant results.
4. Discussion
a. How were treated the patients with CIS? What could we do to decrease the rate of a future MS diagnosis?
Author Response
Thank you for your comments and notes, we tried to answer your questions and corrected the notes.

Reviewer 2 Report (Previous Reviewer 3)
This is a practically important original article indicates the possibility to conversion of CIS in MS and the clinical differences between CSI and MS However, it is necessary to reexamine the research method etc. in several respects.1. Title The title should be indicate in few words the aim of the study. Is it an association study between risk factors?
2. Abstract
Please, write in full words the acronym CSF 3. Introduction The originality of this study is clear. The aim is well explained. Nevertheless, the introduction is poor of information about clinical features and different therapeutic option (pharmacologic and rehabilitation) in MS patients. I suggest to read "Calafiore D et al. Efficacy of Virtual Reality and Exergaming in Improving Balance in Patients With Multiple Sclerosis: A Systematic Review and Meta-Analysis. Front Neurol. 2021 Dec 10;12:773459." to help to better define the different therapeutic options and the impact on disability of MS.
3. Methods, Data source and study participants
Line 66: delete the double brackets
The clinical examination include different clinical outcome as fatigue, generalized weakness, pain, decreased muscle strength, abnormal reflexes, muscle tone abnormalities, vertigo, focal symptoms, cranial nerves dysfunction, pathological reflexes, Babinski‘s reflex, Rossolimo‘s reflex, diminished sense of vibration, proprioception and superficial sensations, ataxia, imbalance, urinary incontinence and retention. How have the Neurologist assessed the different outcome? Please, specify this point (eg. Have you used anmy fatigue scale?). Moreover, it would be desirable to specify the model and MRI tecnique
Have you data about other anthropometric data? (weigth, height, BMI, comormidities?) 6. Statistical analysis, multivariate logistic regression analysis Model 3 and, stratified analyses should add the BMI and comorbidities (if available) 7.Discussion
The limitation of the study should be improve. The absence of other possible confounder factors, the absence of a clera MRI protocol can make lack the results of the research.
Best Regards
Author Response
Thank you for your comments and notes, we tried to answer your questions and corrected the notes.

Reviewer 3 Report (New Reviewer)
Overall I would like to congratulate the authors for their work, it is an interesting article. However, there are some suggestions that I would like to offer:
1) Language editing of the text for clarity purposes;
2) In line 45 and in other lines of the manuscript the authors use the term “paraclinical tests”. This term is used more frequently in British English, is not commonly used in recent scientific publications, and should be replaced.
3) The authors should improve the connection along the introduction paragraph as well as add more information and references about previous publications that also had a similar study protocol in terms of the semiological parameters analyzed and complementary tests used. Especially content from lines 48 to 53 from the introduction needs improvement. Why is this information relevant to the article? The authors do not mention pharmacological treatment or CIMT in the results or in the discussion. The authors should also improve writing from lines 54 to 57 since this is an essential part of the introduction. Please specify which other countries you are mentioning on line 55. The description of the goals of the study should also be improved. What does this study add new? Why is it relevant?
4) On the materials and methods section, the authors should specify why this particular study protocol was chosen, especially regarding inclusion and exclusion criteria. In general, was this study protocol based on other studies – if yes, the authors should add references, was this protocol created by the authors, and in other words, why were these specific parameters analyzed? The clinical questionnaire used should be added to the supplementary material.
5) The authors should specify whether the Hospital in the study was a public or private one. The authors should also indicate the institutional review board number (IRB) of the study with the term IRB followed by the number of the approval.
6) Why was this specific date of following-up chosen? It is a limitation of the study that it reflects data from almost 2 years ago, which should be mentioned in the discussion. How did COVID affect the following-up of the patients or the diagnosis of MS? There are some recent studies correlating an increased number of MS relapses or even MS cases with COVID. Since this study is recent it should at least cite a few articles on this matter in the discussion. The last two years of pandemics are relevant and different conclusions could be drawn if they were included in the study. Please refer to his article for more information:
https://bmcneurol.biomedcentral.com/articles/10.1186/s12883-022-02590-9#:~:text=Earlier%20case%20studies%20%5B6%2C7,11%2C12%2C13%5D.
7) On line 88 of the methods the authors mention that a “broad differential diagnosis was performed”. Please detail this section more. It is also unclear in this paragraph how many years of following up the patients had. Please specify at least the mean years of following up with SD.
8) On the statistical analysis section, more detailing of the methods used should be done. Which was the power of the study? Which tests were used to calculate it? How was the normality of the distribution analyzed? If the distribution was normal, why did the authors use non-parametric tests? Where are the results of these tests? Were the assumptions of logistic regression met, such as linearity, independence of errors, and multicollinearity? Which method of regression was used? On the tables authors should display CI for all the analysis, and for the logistic regression, at least the authors should report beta values, standard errors, R2, and goodness of fit statistics.
9) In the discussion, on line 220 authors should specify which available literature they refer to. The authors should also improve the connection of ideas in the discussion, add limitations mentioned above in the study about the period of the study, external validity of the study, cite more relevant articles regarding study protocol and the parameters the authors chose to analyze, as well as MS vs COVID.
Author Response
Thank you for your comments and notes, we tried to answer your questions and corrected the notes.

Round 2
Reviewer 1 Report (Previous Reviewer 2)
Satisfactory
Author Response
We coorected according reviewer remarks.
Reviewer 2 Report (Previous Reviewer 3)
Dear Authors,
the revision process has improved the quality of the manuscript. At the light of my reading, the manuscript is suitable for the publication.
Best Regards
Author Response
We coorected according reviewer remarks.

Reviewer 3 Report (New Reviewer)
The authors have responded to the suggestions accordingly.
Author Response
We coorected according reviewer remarks.

This manuscript is a resubmission of an earlier submission. The following is a list of the peer review reports and author responses from that submission.
Round 1
Reviewer 1 Report
Dear author,
I found this article very interesting and I would suggest following:
- It would be much more interesting and valuable that you include EDSS in evaluation, as potential risk factor for conversion from CIS to MS. Now it is suitable to leave it like this.
- Age range to 74 years, is unusual for demyelinating disease. Did you do revision of CIS diagnosis by MS experts, before including this patients in study? Older patients could have another differential diagnosis rather than CIS.
- After which average period 49 patients converted to MS? And how long non-MS patients (n=64) were followed? This should be added in results.
- If I can notice, there is high % of unspecified MRI lesions in CIS cohort, how do you explain that? This should be added in discussion.
- How do you explain obtained results of specific MRI lesions (p=0.055) that had no predictive value when predicting CIS conversion to MS? This should be added in discussion.
Best regards
Author Response
Dear Reviewer,
Thank you for your comments. We found them accurate and valuable. These would be our answers.
- It would be much more interesting and valuable that you include EDSS in evaluation, as potential risk factor for conversion from CIS to MS. Now it is suitable to leave it like this.
Since the data was obtained from medical records with patients hospitalized with G37.8 and G37.9 diagnoses for the first time, not G35.0 (multiple sclerosis), the EDSS score was not evaluated for these patients, therefore we haven’t included it.
- Age range to 74 years is unusual for demyelinating disease. Did you do revision of CIS diagnosis by MS experts, before including this patients in study? Older patients could have another differential diagnosis rather than CIS.
This age is unusual but occurs sometimes (recently there was a 71 year patient diagnosed with MS in LUHS Kaunas Clinics). This happens if the patient is hospitalized due to other reasons than suspected MS or simply the patient does not seek medical help for a long time. The CIS diagnosis was done by certified neurologists of LUHS Kaunas Clinics and a broad differential diagnosis was performed. Only 4 of our patients had been diagnosed with other diagnoses than CIS or MS and are mentioned in the results.
- After which average period 49 patients converted to MS? And how long non-MS patients (n=64) were followed? This should be added to the results.
The time during which the patient converted from CIS to MS was not obtained, therefore we cannot include this information. All included patients were followed for conversion to MS until 1st January 2020 regarding the data collected retrospectively.
- If I can notice, there is high % of unspecified MRI lesions in CIS cohort, how do you explain that? This should be added in discussion.
According to the descriptions of MRIs made by the radiologists, lesions were divided into three groups: specific to MS, other CNS lesions demonstrated by MRI, unspecified CNS lesions demonstrated by MRI. By unspecified MRI lesions it was meant that it must be differentiated and the radiologist could not rule out either demyelinating, either angiopathic or vasculitic lesions. The same patient could have had either one type of lesions or all three of them. Our study was limited due to its’ retrospective nature meaning that we were unable to access the actual MRI scan but the description of it was evaluated. Therefore two subjective interpretations had to be done: first the evaluation of the scan by the radiologist and the second by the authors of the study interpreting the description made by the radiologist. Moreover, MRI brain lesion evaluation in Lithuania is not defined by a specific protocol. This leaves more space to the subjective interpretation of the MRI scan. Added to the disscusion.
- How do you explain obtained results of specific MRI lesions (p=0.055) that had no predictive value when predicting CIS conversion to MS? This should be added in discussion.
The same patient could have had either one type of lesions or all three of them. The majority of patients included in the study (n=86) had specific MS MRI lesions, however only 40 patients were observed to have unspecified CNS lesions. The fact that the majority of patients had specific MS MRI lesions, but only 35% of the patients converted to MS brings down the predictive value of specific MS MRI lesions. This could also be explained by the subjective interpretation of MRI by the radiologist and the description of it by the authors. Added to the disscusion.
Best regards
Reviewer 2 Report
1) Abstract. Provide ‘‘OCBs’’ and ‘‘BAEP’’ descriptions.
2) Does the diagnosis of CIS and MS was done by at least two board-certified neurologists?
3) How was assessed the power of the study?
4) Why were the patients divided into three age groups? Please provide a reference for that.
5) Please, provide Spearman correlation graphs as supplementary material.
6) How was the data distributed?
7) How was the model of the multivariate logistic regression done? How were the variables chosen?
8) Please, it is advised to provide a better description of how were the clinical characteristics determined. Was used questionnaires? Was used any specific scale?
9) Table 2. It would be better to do only two columns instead of four. It is difficult to understand.
10) What does this study bring new to the literature?
NEW IDEAS THAT WOULD GREATLY IMPACT THE QUALITY OF THE MANUSCRIPT
i) A figure showing the cumulative number of patients diagnosed with MS throughout the years.
ii) A figure showing the importance of each factor for the development of MS. In it, the size of the circle represents the strength related to that factor. This would be like a summary figure.
Author Response
Dear reviewer,
Thank you for your comments and the profound analysis of the article. These are the answers to your points.
1) Abstract. Provide ‘‘OCBs’’ and ‘‘BAEP’’ descriptions.
Provided in the abstract.
2) Does the diagnosis of CIS and MS was done by at least two board-certified neurologists?
The diagnoses were done by three board-certified neurologists.
3) How was assessed the power of the study?
Power of the study was estimated using one sample T-test.
4) Why were the patients divided into three age groups? Please provide a reference for that.
In this study, patients who were diagnosed with CIS age ranged broadly from 18 to 73 years, so it was difficult to evaluate the data without dividing patients into three separate groups, since some findings (like MRI detected lesions or pathological Brainstem Electric Response Audiometry test findings) are related to patients age (reference: Grose JH, Buss E, Elmore H. Age-Related Changes in the Auditory Brainstem Response and Suprathreshold Processing of Temporal and Spectral Modulation. Trends Hear. 2019 Jan-Dec). Also, writers wanted to find correlations between patients' age and their symptoms/ paraclinical findings that were characteristic to a certain age group.
5) Please, provide Spearman correlation graphs as supplementary material.
It’s included in the supplementary material.
6) How was the data distributed?
Data was distributed normally with an exemption of the sum of related signs and symptoms, which was not normally distributed.
7) How was the model of the multivariate logistic regression done? How were the variables chosen?
Factors related to subsequent MS diagnosis were selected using univariate analysis. Variables that had a significant univariate test were selected as a candidate for the multivariate analysis. We base this on the Wald test from logistic regression and p-value cut-off point of 0.05. All selected variables then were used for multivariate analysis.
8) Please, it is advised to provide a better description of how were the clinical characteristics determined. Was used questionnaires? Was used any specific scale?
Since this study is a retrospective data analysis, writers analyzed data obtained from medical records. The most common clinical characteristics of multiple sclerosis are - fatigue, generalized weakness, pain, decreased muscle strength, abnormal reflexes, muscle tone abnormalities, vertigo, focal symptoms, cranial nerves dysfunction, pathological reflexes, Babinski‘s reflex, Rossolimo‘s reflex, diminished sense of vibration, proprioception and superficial sensations, ataxia, imbalance, urinary incontinence and retention. Writers analyzed medical records, including the anamnesis, neurological examination report and were looking for symptoms that are mentioned above.
9) Table 2. It would be better to do only two columns instead of four. It is difficult to understand.
Added to the newest version.
10) What does this study bring new to the literature?
To our knowledge no other trials regarding Lithuanian population of CIS patients and their conversion to MS were performed. This study contains data from 5 years and gives a wide understanding of what are the most important features when determining if a CIS patient should be monitored even more closely in regard of developing a second demyelinating attack. Moreover, certain prognostic factors of CIS conversion to MS could aid in a more efficient diagnostic process and earlier prescription of the disease modifying therapy.
NEW IDEAS THAT WOULD GREATLY IMPACT THE QUALITY OF THE MANUSCRIPT
i) A figure showing the cumulative number of patients diagnosed with MS throughout the years.
We have not included the dates of MS diagnosis in our database, therefore we can not supply our reviewers with this information.
ii) A figure showing the importance of each factor for the development of MS. In it, the size of the circle represents the strength related to that factor. This would be like a summary figure.
Since there are only two statistically significant results in prognosing CIS conversion to MS, would it still be usefull?
Best regards
Reviewer 3 Report
I have read the paper with great care, and I found it very interesting in the field of neurological diseases. The paper is well written, the methods are clearly described and the discussion is coherent with the results. Nevertheless, there are some critical issues to clarify:
Introduction: The authors describes the sing and symptoms of MS, but there are very lack information about the pharmacological and rehabilitation therapy. Please stress more this point, adding these references:
1) Calafiore, D, Invernizzi, M, Ammendolia, A, Marotta, N, Fortunato, F, Paolucci, T, Ferraro, F, Curci, C, Ćwirlej-Sozańska, A, de Sire, A. (2021). Efficacy of Virtual Reality and Exergaming in Improving Balance in Patients With Multiple Sclerosis: A Systematic Review and Meta-Analysis. Frontiers in Neurology. 12. 773459. 10.3389/fneur.2021.773459.
2) Solaro, C., de Sire, A., Messmer Uccelli, M., Mueller, M., Bergamaschi, R., Gasperini, C., Restivo, D.A., Stabile, M.R., & Patti, F. (2020). Efficacy of levetiracetam on upper limb movement in multiple sclerosis patients with cerebellar signs: a multicenter double‐blind, placebo‐controlled, crossover study. European Journal of Neurology, 27.
3) de Sire, A., Mauro, A., Priano, L., Baudo, S., Bigoni, M., & Solaro, C. (2019). Effects of Constraint-Induced Movement Therapy on upper limb activity according to a bi-dimensional kinematic analysis in progressive multiple sclerosis patients: a randomized single-blind pilot study. Functional neurology, 34 3, 151-157 .
4) de Sire, A., Bigoni, M., Priano, L., Baudo, S., Solaro, C., & Mauro, A. (2019). Constraint-induced movement therapy in multiple sclerosis: Safety and three-dimensional kinematic analysis of upper limb activity. A randomized single-blind pilot study. NeuroRehabilitation.
Methods: nothing to declare
Results: nothing to declare
Discussion: please, add a clinical relevance of the paper in the field of early diagnosis of MS in CIS framework
Author Response
Dear reviewer,
Thank you for your comments, we found them useful.
These would be the answers to your points:
- References were added to the introduction.
- Relevance of the paper was added to the disscusion. ''
To our knowledge no other trials regarding Lithuanian population of CIS patients and their conversion to MS were performed. This study contains data from 5 years and gives a wide understanding of what are the most important features when determining if a CIS patient should be monitored even more closely in regard of developing a second demyelinating attack. Moreover, certain prognostic factors of CIS conversion to MS could aid in a more efficient diagnostic process and earlier prescription of the disease modifying therapy."
Best regards
Round 2
Reviewer 2 Report
1) OK
2) OK
3) This explanation should be written in the manuscript.
‘’Power of the study was estimated using one sample T-test.’’
4) This explanation should be written in the manuscript.
‘‘In this study, patients who were diagnosed with CIS age ranged broadly from 18 to 73 years, so it was difficult to evaluate the data without dividing patients into three separate groups, since some findings (like MRI detected lesions or pathological Brainstem Electric Response Audiometry test findings) are related to patients age (reference: Grose JH, Buss E, Elmore H. Age-Related Changes in the Auditory Brainstem Response and Suprathreshold Processing of Temporal and Spectral Modulation. Trends Hear. 2019 Jan-Dec). Also, writers wanted to find correlations between patients' age and their symptoms/ paraclinical findings that were characteristic to a certain age group.’’
5) The reviewer was not able to access supplementary files. Please, if possible, provide it again.
6) OK
7) OK
8) Still need to be improved the description in text because if no standard questions were asked. A pilot study should be done.
9) OK
10) OK
Author Response
3) Is written now under statistical analysis section.
4) Is written now under materials and methods section.
5) Sorry for the inconvenience.
8) Added: "Regarding the retrospective nature of the study clinical variables were acquired from medical documentation, including patient complaints and neurological examination performed by neurologists of LUHS Kaunas Clinics." to the materials and methods section.
Reviewer 3 Report
After the revision, the paper is suitable for the publication
Best regards
Author Response
Thank you very much.